# Proteomic Profiling Revealed Mitochondrial Dysfunction in Photoreceptor Cells under Hyperglycemia

**DOI:** 10.3390/ijms232113366

**Published:** 2022-11-01

**Authors:** Christie Hang-I Lam, Jimmy Ka-Wai Cheung, Dennis Yan-Yin Tse, Thomas Chuen Lam

**Affiliations:** 1School of Optometry, The Hong Kong Polytechnic University, Kowloon, Hong Kong; 2Centre for Eye and Vision Research (CEVR), 17W Hong Kong Science Park, Hong Kong; 3Research Centre for SHARP Vision (RCSV), The Hong Kong Polytechnic University, Kowloon, Hong Kong

**Keywords:** diabetic retinopathy, SWATH-MS, mitochondrial dysfunction

## Abstract

Diabetic retinopathy (DR) was identified as a leading cause of blindness and vision impairment in 2020. In addition to vasculopathy, DR has been found to involve retinal neurons, including amacrine cells and retinal ganglion cells. Despite possessing features that are susceptible to diabetic conditions, photoreceptor cells have received relatively little attention with respect to the development of DR. Until recently, studies have suggested that photoreceptors secret proinflammatory molecules and produce reactive oxygen species that contribute to the development of DR. However, the effect of hyperglycemia on photoreceptors and its underlying mechanism remains elusive. In this study, the direct effect of high glucose on photoreceptor cells was investigated using a 661w photoreceptor-like cell line. A data-independent sequential window acquisition of all theoretical mass spectra (SWATH)-based proteomic approach was employed to study changes induced by high glucose in the proteomic profile of the cells. The results indicated that high glucose induced a significant increase in apoptosis and ROS levels in the 661w cells, with mitochondrial dysfunction among the major affected canonical pathways. The involvement of mitochondrial dysfunction was further supported by increased mitochondrial fission and reduced mitochondrial bioenergetics. Collectively, these findings provide a biological basis for a possible role of photoreceptors in the pathogenesis of DR.

## 1. Introduction

Diabetic retinopathy (DR) is one of the leading causes of blindness in the working-age population [1], reported as the cause of one in 39 cases worldwide [2]. This ocular disease not only leads to reduced quality of life and increased mortality of sufferers but also results in a considerable global economic burden. The prevalence of DR is increasing rapidly; the number of DR patients is expected to increase from 7.7 million in 2010 to 14.6 million by 2050 in the U.S. alone (National Institutes of Health National Eye Institute data: https://nei.nih.gov/eyedata/diabetic#5, accessed on 23 June 2021). Therefore, disease prevention and the development of effective treatments are of considerable importance.

Although DR is commonly associated with microvascular abnormalities, leading to vascular endothelial dysfunction (e.g., compromised blood–retinal barrier, vessel leakage, and oedema), diabetes also affects the neural units of the retina, causing gradual neurodegeneration, gliosis, and neuroinflammation. However, emerging evidence has suggested that the functional and structural alterations involved in the neuroretina of the diabetic eye may be the earliest manifestation of DR and could be used to predict the progression of angiopathy [3,4,5,6]. For instance, progressive thinning of the inner retina has been reported in diabetic patients and experimental mouse models of diabetes with minimal or even no change in the retinal vasculature [7]. Delayed implicit times of multifocal electroretinogram signals have also been observed in diabetic patients prior to any detectable signs of angiopathy [8].

Among the various types of retinal neurons, the importance of photoreceptor cells has received limited recognition as a factor in the development of DR. However, they possess some features that are susceptible to diabetic conditions and may contribute to the pathogenesis of DR. For instance, subnormal changes in proteins that are critical to photoreceptor functions, such as rhodopsin and transducin, have been observed in the diabetic retina without manifested angiopathy [9,10]. In addition, insulin is closely related to the function of photoreceptor cells [11,12,13], such that both absolute and relative deficits in insulin, which are crucial features of type 1 and type 2 diabetes, respectively, could influence photoreceptor cells. Recent studies have revealed that the photoreceptor cells could secret proinflammatory molecules, leading to a direct and indirect impact on the retinal vasculature, further highlighting the contribution of photoreceptor cells to the pathogenesis of DR [[14],[15],[16],].

However, the existing literature documenting the degeneration of photoreceptor cells under diabetic stress remains controversial [6,17,18,19,20]. In this regard, the aim of the present study was to investigate the effect of high glucose on photoreceptors using a 661w photoreceptor-like cell line in order to advance knowledge on the mechanisms of DR and help to develop a novel treatment regime to preserve the vision of diabetic patients.

## 2. Results

### 2.1. Validation of Cone Photoreceptor-Specific Markers in 661w Cells

The 661w cell line is a mouse photoreceptor-derived cell line that is immortalized by the expression of SV40 T antigen under the control of human interphotoreceptor retinoid-binding protein promoter [21]. Although, morphologically, these cells exhibit processes that are generally observed in neuronal cells and do not form outer segment-like membranes as observed in photoreceptors, it was demonstrated that they display blue LED light-induced damage, similar to that observed in primary retinal photoreceptors [22]. Previous studies have reported the expression of both rod and cone photoreceptor cell markers in this cell line, such as blue and green opsin, transducin, rhodopsin, arrestin, RDS/peripherin, and phosducin [23,24,25], whereas an earlier study reported a lack of rod and cone photoreceptor phenotypes [21]. In the present study, we first determined whether the 661w cells are a suitable cell line to study photoreceptor physiology. The immunohistochemistry results revealed that 661w cells displayed positive staining with red/green opsin (Figure 1a), which is a cone photoreceptor-specific marker. In addition, the 661w cells exhibited positive immunoreactivity toward rhodopsin, a marker of rod photoreceptors (Figure 1b). The expression of opsin and rhodopsin in 661w cells was confirmed by Western blot (Figure 1c).

### 2.2. High Glucose Induced Changes in Cell Viability, Apoptosis, and ROS Levels in 661w Cells

The effects of high glucose on cell viability and apoptosis of 661w cells were evaluated after 48 h incubation. Trypan blue assay revealed a lower but statistically insignificant decrease in cell viability in 661w cells incubated in high glucose compared to the normal glycemic control (normal glycemic control: 83.1 ± 4.2% vs. high glucose: 74.3 ± 9.3%, *p* = 0.42). In contrast, results from the annexin V apoptosis assay showed that high glucose significantly increased cell apoptosis by approximately 1.4-fold (Figure 2a–c) compared to the normal glycemic control (independent sample *t*-test, *t* = −7.775, *p* < 0.001). 661w cells incubated in high glucose also exhibited a significant increase in CM-H2DCFDA fluorescein, indicating an increase in ROS levels (1.7-fold) compared to the normal glycemic control after 48 h incubation (independent sample *t*-test, *t* = −3.598, *p* < 0.01) (Figure 2d).

### 2.3. Changes in Proteomic Profiling of the 661w Cells under High-Glucose Conditions

To investigate the changes in the proteomic profile and the underlying molecular events in 661w cells after acute exposure to a high-glucose environment, a data-independent sequential window acquisition of all theoretical mass spectra (SWATH)-based proteomic analysis was performed after the 48 h incubation. The dataset was loaded onto the OneOmics cloud−based platform (SCIEX) for SWATH quantification, where a total of 2802 proteins (at 1% FDR and with more than one pep-tide per protein) were obtained. After filtering out the proteins with confidence ≥65%, 981 proteins were left. Of these proteins, 625 met the cutoff criteria of differential expression (fold-change ≥1.5 or ≤0.67, *p* < 0.05, confidence ≥65%), of which 313 were upregulated and 312 were downregulated. Figure 3 shows the volcano plot illustrating the distribution of the 2802 proteins in our dataset, with those highlighted indicating the 981 proteins. The protein IDs of the 981 proteins were converted to their gene name using the Uniprot database and uploaded to the Ingenuity^®^ Pathway Analysis (IPA) online bioinformatics application in order to reveal their significant canonical pathways. A total of 102 significant pathways was revealed by pathway analysis using the IPA. The top five significant pathways and their associated *p*-values and z-scores are shown in Table 1, including (1) the protein ubiquitination pathway, (2) mitochondrial dysfunction, (3) oxidative phosphorylation, (4) epithelial adherens junction signaling, and (5) actin cytoskeleton signaling. The top tox list analysis of IPA also revealed that mitochondrial dysfunction ranked as the highest toxicity-related biological process in the dataset.

### 2.4. Validation of Proteins in Mitochondrial Dysfunction Using Western Blot

Based on the results obtained with IPA, 26 proteins involved in mitochondrial dysfunction were found to be differentially expressed in the 661w cells incubated with high glucose compared to the normal glycemic control (Table 2). Four of these were selected for further validation using Western blot, and the results confirmed the upregulation of oxoglutarate dehydrogenase (Ogdh) and the downregulation of superoxide dismutase 2 (Sod2), synuclein alpha (Snca), and peroxiredoxin 5 (Prdx5) (Figure 4).

### 2.5. High Glucose Induces Mitochondrial Dysfunction in 661w Cells

The functional impact of high glucose on mitochondrial bioenergetics of the 661w cells was then evaluated by measuring the real-time oxygen consumption rate of the cells using a Seahorse XF Cell Mito Stress assay. Several chemicals that target the electron transport chain (ETC) complexes were serially injected throughout the assay. These chemicals included oligomycin (a complex V inhibitor), carbonylcyanide-4-(trifluoro-methoxy) phenylhydrazone (FCCP, a mitochondrial uncoupler), and a mixture of rotenone (a complex I inhibitor) and antimycin A (a complex III inhibitor). The result was further divided into key parameters of mitochondrial function, as illustrated in Figure 5a,b. We observed significant reduction in ATP production (normal glycemic control: 9.86 ± 0.38 vs. high glucose: 7.62 ± 0.84; independent sample *t*-test, *t* = 2.427, *p* = 0.041) and maximal respiration (normal glycemic control: 14.33 ± 0.89 vs. high glucose: 10.52 ± 0.86; independent sample *t*-test, *t* = 3.073, *p* = 0.015) in the 661w cells after incubation in high glucose compared to the normal glycemic control. Despite a tendency of reduced basal respiration and spare capacity in the 661w cells after incubation with high glucose, the result did not reach statistical significance.

### 2.6. High Glucose Induces Mitochondrial Network Fragmentation and May Be Associated with Increased Cell Apoptosis

The effects of high glucose concentration on the mitochondrial morphology of the 661w cells were also assessed. Following 48 h of incubation, the mitochondria of the cells were stained with MitoTracker^TM^ Green FM (MTG), which is a mitochondrial membrane potential independent dye. The mitochondrial morphology of the cells was captured with live cell imaging using a confocal microscope, and these images were analyzed with ImageJ to calculate the form factor (FF) and aspect ratio (AR), as described in [26]. FF indicates the mitochondrial length and degree of mitochondrial branching. An FF value of one reflects a circular, unbranched mitochondrion, whereas a higher FF value implies a longer and more branched mitochondrion. In contrast, AR is a measure of mitochondrial circularity, whereby a value of one indicates a perfect circle, with the value increasing with elongated and elliptical mitochondria. The results shown in Figure 6 reveal that incubation of the 661w cells in high glucose resulted in a significantly reduced FF (normal glycemic control: 4.82 ± 0.42 vs. high glucose: 2.95 ± 0.40; Mann–Whitney U test, U = 111, *p* = 0.003) and AR (normal glycemic control: 2.44 ± 0.08 vs. high glucose: 1.92 ± 0.29; Mann-Whitney U test, U = 119, *p* < 0.001) values when compared with the normal glycemic control. These results suggest that the mitochondria of 661w cells incubated in high glucose had fewer branches and became more circular in shape, implicating an increase in mitochondrial network fragmentation compared to the normal glycemic control.

The changes in mitochondrial morphology under high glucose were found to be accompanied by an increased level of mitochondrial fission 1 (Fis1), a protein that could mediate mitochondrial fission [27]. The expression of Fis1in the 661w cells under high glucose was found to be increased by approximately 1.35-fold (independent sample *t*-test, *t* = −6.204, *p* = 0.003) compared to the normal glycemic control (Figure 6e,f). Interestingly, the molecular size of Fis1 was found to be approximately 17 kDa, whereas the Western blot result detected a band at around 68 kDa. Fis1 is known to interact with several proteins, such as B-cell receptor-associated protein 31 [28], dynamin-related protein 1 [29], family members of PEX11 [30], and TBC1 domain family member 15 [31]. The differing molecular size of the detected band may be the result of complex formation with mitochondrial elongation factor 1 (Mief1), a 51 kDa protein [32].

To investigate whether the high-glucose-induced increase in cell apoptosis in 661w cells was due to the mitochondrial-mediated cell death pathway, the extent of cytochrome c translocation from mitochondria to cytosol was evaluated with Western blotting after subcellular fractionation. The 661w cells exposed to high glucose for 48 h showed a significant increase (fold change = 2.53; independent sample *t*-test, *t* = −3.363, *p* = 0.028) in cytochrome c levels in the cytosolic protein fraction (Figure 6g,h) when compared to the normal glycemic control.

## 3. Discussion

Photoreceptor cells, one of the major cell classes in the retina, have been suggested to play a crucial role in the development of DR by secreting proinflammatory molecules to affect other retinal cell types [[14],[15],[16],]. However, there are controversial reports with respect to the change or degeneration of photoreceptor cells under diabetic stress across different animal models [6,17,18,19,20]. How this cell type is affected during the course of diabetes remains to be elucidated. The results of the current study show that, although not leading to a significant increase in cell death, exposure to high glucose could induce cell apoptosis in the 661w photoreceptor-like cell line after 48 h incubation.

Neurodegeneration has been suggested to be an early event in DR that could lead to irreversible vision loss [3,4,5,6]. In order to gain mechanistic insight into how diabetes-induced neurodegeneration develops, the proteomic changes of 661w cells after exposure to high glucose was studied by employing a data-independent SWATH-based proteomic analysis coupled with IPA pathway analysis. The results revealed 625 differentially expressed proteins and 102 significant canonical pathways, reflecting the multifactorial nature of DR. Among these significantly enriched pathways, mitochondrial dysfunction was found to rank second among the top significant pathways and the top toxicity-related biological process.

Mitochondrial dysfunction has been implicated in several retinal pathological conditions, including Leber hereditary optic neuropathy, age-related macular degeneration, and glaucoma [33,34]. Although the major function of mitochondria is to produce ATP through oxidative phosphorylation, it has emerged that mitochondria are also involved in other processes critical for the preservation of cell integrity and survival. These functions include regulation of mitochondrial dynamics (fission and fusion) [35,36,37]; maintenance of calcium homeostasis [38,39,40] and nucleotide metabolism [41,42]; and biosynthesis of amino acids [43,44], cholesterol, and phospholipids [45]. Thus, functional and structural impairment of mitochondria could markedly compromise tissue and cellular function.

In addition, mitochondrial dysfunction has been associated with the onset of Type 2 diabetes and the development of insulin resistance and diabetes-related complications [46,47,48]. Impaired mitochondria have also been implicated in the death of various retinal cell types, including retinal endothelial cells [49], pericytes [50], and Mϋller cells [51], under simulated hyperglycemia. Photoreceptors exhibit remarkably high metabolic activity, especially at the inner segment [52], although their reserve capacity for mitochondrial oxidative phosphorylation has been shown to be somewhat limited [53]. Based on these unique features compared with other retinal cells, we speculate that photoreceptor cells may be more readily affected by diabetes-induced mitochondrial dysfunction. To understand how mitochondria of photoreceptor cells are affected under high-glucose concentration, the mitochondrial bioenergetic characteristics and morphology of the 661w cells were further investigated under elevated glucose.

The effect of high glucose on the mitochondrial bioenergetics of 661w cells was assessed by evaluating the real-time oxygen consumption rate using a Seahorse XF analyzer. It has been suggested that quantifying the oxygen consumption rate could better reflect the impairment of the electron transport chain than measuring other intermediates of cellular respiration, such as ATP and nicotinamide adenine dinucleotide [54]. The results showed that after 48 h of incubation, the 661w cells incubated in high glucose exhibited a significant reduction in ATP production and maximal respiration levels compared to the normal glycemic control. A decrease in maximal respiration has been suggested to be an indicator of mitochondrial dysfunction [55]. Our results demonstrate that high glucose led to compromised mitochondrial function in 661w cells, which likely reduces the efficacy of the ATP synthesis machinery of the mitochondria to deal with an unusually high energy demand. The ATP production evaluated by this assay is based on the changes in OCR after the addition of oligomycin, an ATP synthase inhibitor. It remains uncertain whether the ATP production paradigm will shift from a more effective oxidative phosphorylation to less effective glycolysis under high-glucose conditions to maintain ATP levels to support cell survival.

The results revealed an increase in mitochondrial network fragmentation in the 661w cells after 48 h incubation in high glucose compared to the normal glycemic control. Correspondingly, reduced FF and AR values were demonstrated. The results suggest that the mitochondria of the 661w cells became less branched and elongated and more circular in shape, suggesting a reduction in mitochondrial efficiency. Mitochondrial network fragmentation has been suggested to be an inducer of ROS production, which may occur via a feed-forward mechanism [56]. Increased mitochondrial fragmentation has also been demonstrated in other retinal cell types under incubation with high glucose [49,50,51], as well as in the vascular cells of the retinal capillaries of diabetic rats [57].

The increased mitochondrial network fragmentation under high glucose was found to accompany an upregulation of Fis1. Fis1, which is present in the mitochondrial outer membrane, has been found to mediate mitochondrial fission and be involved in apoptotic and mitophagic pathways [27,29,58,59]. Western blot analysis detected a predominant signal at around 68 kDa instead of at 17 kDa, which is the usually documented molecular size of Fis1. This suggests that Fis1 has a tendency to bind with other proteins in 661w cells, among which Mief is believed to be a possible candidate [32]. In addition, the results reveal that the protein level of cytochrome *c* in the cytosol fraction was significantly increased in 661w cells under high-glucose conditions. This result implies that high glucose led to an increase cytochrome c translocation in 661w cells, a key initial step that signals cell apoptosis [60]. However, how upregulation of Fis1 and its interaction with other proteins affects mitochondrial dynamics and apoptosis in 661w cells remains to be elucidated.

Recently, it has been shown that protein levels of several subunits of the ETC complexes, particularly complex I, were increased when incubating the primary cardiac microvascular endothelial cells (RCMVECs) obtained from diabetes-susceptible Goto-KakiZaki (GK) rats under high-glucose conditions [61]. Compared to the normal glucose control, phorbol 12-myristate 13-acetate-induced ROS production was observed in the high-glucose-treated GK RCMVECs. On the contrary, our MS results suggest decreased levels of several ETC complex subunit proteins, such as Ndufa8, Ndufs1, Ndufas2, Ndufas3, and Ndufas4 in complex I, in 661w cells under high-glucose conditions compared to the normal glucose control. We also observed increased basal ROS levels in the high-glucose-treated 661w cells. Whereas the previous study used RCMVECs from GK rats (i.e., a polygenic model of type 2 diabetes) and incubated the cells under high-glucose conditions for 13 weeks, in our study, we adopted 661w cells (i.e., a photoreceptor-like cell line) and exposed the cells to high-glucose conditions for only two days. The difference in the cell types and methodologies (e.g., the duration of high-glucose exposure and the approach to measuring ROS levels) may account for the discrepancies. Although ROS production is associated with the activity of mitochondria during ATP production through oxidative phosphorylation, it has also been reported that defective or damaged mitochondria and ETC complexes could alter ROS levels [62,63,64]. For instance, inactive complex III has been implicated in increased ROS levels in the retina of STZ-induced diabetic mice [65]. In addition, complex I deficiency is associated with a broad spectrum of pathologies, including but not limited to Leber hereditary optic neuropathy and Parkinson’s disease, in which ROS overproduction has been reported as one of the critical pathological phenotypes [66,67,68]. It should be noted that ETC complexes often consist of complicated structures, and the functions of individual subunits remain largely unknown. Further studies are warranted to improve understand of the role of ETC complexes and their associated subunits under physiological and pathological conditions.

One of major limitation of this study is that the cells chosen for the experiments are a cell line rather than primary cells; therefore, the effects may not fully replicate the condition of photoreceptor cells subjected to diabetic stress. Nevertheless, as cell lines are inexpensive and easy to handle and render better consistency across samples and more reproducible results than primary cells [69], they are often used as a platform to study biological processes and to test drug metabolism and cytotoxicity [69,70]. Overall, the results of this study suggest the possible involvement of mitochondrial dysfunction in photoreceptor cells under high-glucose conditions. These findings complement the growing body of studies that implicate the involvement of mitochondrial dysfunction in the pathogenesis of DR [49,50,51,57,71].

## 4. Materials and Methods

### 4.1. Cell Culture and Treatment

The 661w cell line, generously provided by Prof. Muayyad Al-Ubaidi, University of Houston, was cultured (passage 21–25) and incubated in 6-well plates at 37 °C in a 5% CO_2_ incubator with Dulbecco’s modified eagle medium (DMEM, Gibco, Grand Island, NY, USA) containing 10% fetal bovine serum (FBS) and 1× antimycotic solution as described previously [72]. When confluency reached 80%, the growth medium was replaced with serum-reduced medium (0.5% FBS) overnight prior to the experiments. The cells were then exposed to 5.5 mM glucose (normal glycemic control) or 55 mM glucose (high glucose) in serum-reduced medium for 48 h.

### 4.2. Immunocytochemistry

The 661w cells were fixed in 4% paraformaldehyde for 10 min at room temperature, permeabilized with 0.1% Triton X-100 for 10 min, and blocked in phosphate-buffered saline (PBS) containing 10% donkey serum for 30 min at room temperature. The cells were then incubated overnight with primary anti-opsin red/green (PA1-9517, dilution 1:250, Invitrogen, Waltham, MA, USA,) or anti-rhodopsin antibody (Invitrogen, PA5-102150, dilution 1:200) at 4 °C. Following incubation, the cells were washed thrice with ice-cold PBS and incubated with donkey-host secondary antibody conjugated with Alexa 647 (ab150107, dilution 1:500, abcam, Boston, MA, USA) or Alexa 488 (A21206, dilution 1:500, Invitrogen) for one hour. DAPI (D1306, Invitrogen) was used to stain the nuclei of the cells. After three washes with ice-cold PBS, the cells were subjected to immunocytochemistry using a confocal laser scanning microscope (LSM800, Zeiss, Oberkochen, Germany) under with a 40× oil objective to capture the confocal micrographs.

### 4.3. Cell Apoptosis Assay

The extent of apoptosis of the 661w photoreceptor-like cell line was assessed using and FITC annexin V/dead cell apoptosis kit (V13242, Invitrogen-Molecular Probes, Eugene, OR, USA) and flow cytometry. After 48 h incubation, the cells were harvested and washed in cold PBS. The cells were then collected after centrifugation at 210× *g* for 6 min at 4 °C and resuspended in 100 μL annexin-binding buffer. Volumes of 2 μL of FITC annexin V and 1 μL of 100 μg/mL propidium iodide (PI) were added to the cell suspension, and the mixture was incubated for 15 min. After incubation, 400 μL of 1× annexin-binding buffer was added and mixed gently, and the samples were kept on ice until analysis using a BD FACSVia™ flow cytometer (BD Biosciences, Franklin Lakes, NJ, USA). The fluorescence emission was measured at 530 nm and >575 nm.

### 4.4. Cell Viability Assay

The cell viability of the 661w photoreceptor-like cell line was determined by a Trypan blue assay using an Countess^TM^ II FL automated cell counter (Life Technologies Corporation, Bothell, WA, USA). After harvesting the cells, 15 μL of cell suspension was mixed with 15 μL of trypan blue (T10282, Invitrogen) and rested for 30 s before injection to the automated cell counters for analysis.

### 4.5. ROS Level Evaluation

A general oxidative stress indicator, CM-H2DCFDA, was used to examine the change in ROS levels in the 661w cell line. After 48 h incubation, the 661w cells were harvested. The cells pallets were resuspended in 300 μL PBS and divided into two equal portions. One portion of the cells was stained with 50 μL CM-H2DCFDA dye (C6827, Invitrogen) at a working concentration of 10μM, and the second was stained with Hoechst 33342, a nuclear dye, for normalization. The cells were then incubated at 37 °C for one hour and protected from light. Subsequently, the cells were washed with PBS, centrifuged at 210× *g* for 6 min, and lysed with buffer containing 0.75 M HCL and 0.2% Triton-X 100 at room temperature for 30 min in darkness. The supernatants were collected after centrifuging the cell lysate at 21,380× *g* for 5 min, and 100 μL was added to a well of a 96-well plate with a technical replicate. The fluorescence level was measured by a CLARIOstar^®^ microplate reader (BMG LABTECH, Ortenberg, Germany).

### 4.6. Sample Preparation for LC-MS/MS

After 48 h of treatment, each well was rinsed briefly with PBS. The cells were then trypsinized and harvested (normal glycemic control: n = 3, high glucose: n = 6). The cell pellets were collected after centrifugation at 300× *g* and treated with S-Trap SDS protein solubilization buffer containing 10% SDS and 100mM triethylammonium bicarbonate (TEAB) [73]. After sonication on ice for 30 min, the protein concentrations were assessed by a Pierce^TM^ BCA gold protein assay (Thermo Fisher Scientific, Rockford, IL, USA) according to the manufacturer’s guidelines.

The protein concentration was adjusted to 2 μg/μL with S-Trap SDS protein solubilization buffer. Then, 200 mM DTT was added to 25 μL protein solution to afford a final concentration of 20 mM. The samples were then incubated at 95 °C for 10 min and cooled to room temperature. The cysteines were alkylated by adding 400 mM IAA to a final concentration of 40 mM, and the mixture was incubated in the dark for 30 min. Aqueous phosphoric acid (12%) was added at a ratio of 1:10, followed by the addition of a sixfold volume of S-Trap protein-binding buffer to the acidified lysis buffer and thorough mixing. The acidified SDS lysate/S-Trap buffer mixture was transferred into the microcolumn by centrifugation at 4000× *g* twice. The protein was then washed by adding 150 μL S-Trap protein-binding buffer and centrifugation three times at 4000× *g*.

Trypsin was added to the sample at a 1:25 *w*/*w* ratio (enzyme/protein), and the mixture was incubated at 47 °C for one hour. The peptides were eluted with 40 μL of 50 mM TEAB and 0.2% aqueous formic acid. The hydrophobic peptides were recovered by elution with 35 μL 50% acetonitrile in 0.2% formic acid and dried using a speed vacuum centrifuge. The peptide concentration was then assessed by a Pierce^TM^ quantitative colorimetric peptide assay (Thermo Fisher Scientific) and adjusted to a final concentration of 0.5 μg/μL with 0.1% formic acid. An equal amount of peptide was drawn from each sample for each condition and pooled together. A total of 2 μg peptide was injected into the LC−MS for protein identification and to establish a protein ID library with technical replicates.

### 4.7. MS Analysis

All MS data were acquired using a TripleTOF**^®^** 6600 quadrupole time-of-flight (QTOF) mass spectrometer (QTOF, SCIEX, Framingham, MA, USA) with Analyst TF 1.7 software. The ion source was operated with the following parameters: ISVF = 2300; GS1 = 15; CUR = 30; and IHT = 120 with Analyst software (v1.7.1, SCIEX, USA).

For data-dependent acquisition (DDA) a TOF-MS scan over a mass range of 350−1800 *m*/*z* with 250 ms accumulation time was performed, followed by 100−1800 *m*/*z* for MS/MS scans in high-sensitivity mode with 50 ms accumulation time for the top 50 ion candidates per cycle. Ions that exceeded the threshold of 125 cps were counted. The rolling collision energy was selected to trigger collision-induced dissociation.

For SWATH MS-based (SWATH-MS) experiments, the instrument was tuned for a variable isolation window in a looped mode over the mass range of 100 *m*/*z* to 1800 *m*/*z* to scan 100 overlapping variable windows. An accumulation time of 29 ms was set for each fragment ion, resulting in a total duty cycle of 3.0 s.

#### 4.7.1. Protein Identification and Generation of the Protein Library

The DDA data were searched against the Mouse Uniprot database (updated on 13 June 2020, 55,398 entries) using ProteinPilot (v5.0, SCIEX, USA) for protein identification. Trypsin was set as the digesting enzyme, and iodoacetamide was set as the alkylating agent of cysteine residues. A “thorough” search setting with the inclusion of biological modifications was used. Peptide and protein identifications were filtered to obtain a global false-discovery rate (FDR) of 1%.

The DDA files obtained from the technical duplicate of the pooled sample were combined to generate a protein ID library, which identified a total of 4745 proteins and 51,702 peptides at 1% FDR. This combined library was used as the reference library for the MS-based quantitative analysis for the experiment.

#### 4.7.2. SWATH-MS (Label-Free) Quantification

A combined peptide spectral library was generated using the identified peptides, and the corresponding peptide fragment peaks for each peptide were extracted by SWATH Acquisition MicroApp 2.0 in PeakView (v2.2, SCIEX, Framingham, MA, USA). This library and all raw SWATH files were then uploaded to the OneOmics cloud-based online platform for spectral matching, processing, and statistical analysis, with a 1% global FDR threshold set for both protein and peptide levels. Ten peptides per protein and the top six transitions per peptide were used for peak extraction and quantification. RT calibration was performed using an auto-calibration setting, whereby peptides were used across the whole spectrum. Shared peptides and peptides with modifications were excluded from quantification. Extracted ion chromatograms (XICs) over a 5-min extraction window with 75 ppm fragment mass tolerance were applied. After processing, the MS data were normalized based on the Most-Likely-Ratio (MLR) algorithm [74], which considers both the biological and technical replicates and calculates sample scores and measurement weights for downstream ratio analysis. Only proteins passing 1% FDR and with at least two quantifiable peptides were included in the analysis. Proteins were considered to be differentially expressed according to four criteria: (1) *p*-value < 0.05 (unpaired *t*-test), (2) fold change (FC) ≥ 1.50 or ≤0.67 (log_2_ FC ≥ 0.585 or ≤−0.585), (3) confidence ≥ 65%, and (4) repeatability ≥ 0.15.

### 4.8. Bioinformatics Analysis

The IDs of identified proteins were converted into gene names with reference to the Uniprot protein online database for subsequent bioinformatics analysis. Pathway analysis for protein−protein interactions was performed using Ingenuity^®^ Pathway Analysis (IPA). The schematic workflow of quantitative discovery proteomics in 661w cells after 48 h incubation in high glucose vs. the normal glycemic control is shown in Figure 7.

### 4.9. Western Blot Analysis

The 661w cells were washed with PBS and lysed with EB2 lysis buffer, which contained 7 M urea, 2 M thiourea, 30 mM Tris, 2% (*w*/*v*) CHAPS, and 1% (*w*/*v*) ASB14 with protease inhibitor (PI) cocktail (Roche Applied Science, Basel, Switzerland), sonicated for 30 min on ice, and centrifuged at 21,380× *g* for 15 min.

Subcellular fractionation was performed to study cytochrome c release. The 661w cells were washed with PBS and lysed with 0.1% Triton X-100 buffer containing 20 mM HEPES (pH 7.4), 10 mM potassium chloride, 2 mM magnesium chloride, 1 mM EDTA, 1 mM EGTA, 1 mM DTT, and 1× PI cocktail. The lysed mix was centrifuged at 720× *g* for 5 min. The supernatant was collected and centrifuged at 20,000× *g* for 10 min. The supernatant was collected as the cytosolic protein fraction. The remaining cellular pellet was resuspended with TBS/0.1% SDS and centrifuged at 20,000× *g* for 10 min. The supernatant was then collected as the mitochondrial protein fraction.

The protein concentration was determined by a Bio-Rad Bradford protein assays according to the manufacturer’s instruction. A 50 μg aliquot of protein was drawn from each sample and subjected to electrophoresis in 12% sodium dodecyl sulphate (SDS) polyacrylamide gels, after which the protein samples were transferred to polyvinylidene fluoride (PVDF, Bio-Rad Laboratories, Hercules, CA, USA) membranes. Non-specific binding was blocked by incubation with 5% non-fat dried milk for one hour. The blots were incubated overnight at 4 °C with primary anti-SOD2 (sc-133134, dilution 1:500, Santa Cruz Biotechnology, Dallas, TX, USA), anti-α-synuclein (ab212184, dilution 1:1000, abcam), anti-peroxiredoxin 5 (ab180587, dilution 1:1000, abcam), anti-Ogdh (ab137773, dilution 1:1000, abcam), anti-Fis1 (LS-C404988, dilution 1:1000, LSBio, Seattle, WA, USA), anti-cytochrome c (MA5-11674, dilution 1:500, Invitrogen), anti-opsin red/green (PA1-9517, dilution 1:1000, Invitrogen), anti-rhodopsin antibody (PA5-102150, dilution 1:500, Invitrogen), or anti-β-actin (MA5-11869, dilution 1:1000, Invitrogen,) in 5% non-fat dried milk overnight. Blots were washed three times with TBS containing 0.1% Tween (TBS-T) and were then incubated with the secondary antibodies for one hour at room temperature with gentle agitation. After three washes with TBS-T, the membrane was exposed to SuperSignal^TM^ West Pico PLUS chemiluminescent substrate (Thermo Fisher Scientific) for detection of protein signals. Analysis of densitometry was conducted using ImageJ software (v1.53k, National Institutes of Health, Bethesda, MD, USA). The protein signals were normalized to the corresponding β-actin signal to ensure equal loading of protein. The results were then compared to the normal glycemic control to calculate the relative expression.

### 4.10. Confocal Microscopy and Image Analysis for Mitochondrial Morphology

The 661w cells were grown on 35 mm glass-slide-bottomed dishes (ibidi, Martinsried, Germany) and incubated in corresponding experimental conditions for 48 h before confocal microscopy. The 661w cells were incubated at 37 °C in a 5% CO_2_ incubator with 100 nM membrane potential independent dye (MitoTracker^TM^ Green FM [MTG]; Invitrogen-Molecular Probes) for 45 min before imaging and kept in the medium during the imaging processes. The mitochondrial morphology of the cells was captured under different experimental conditions with live cell imaging using a Leica TCS SPE confocal microscope (Leica Microsystems, Nussloch, Germany) with an oil magnification lens at 63×.

MTG was subjected to 488 nm solid-state laser excitation and emission was recorded through a 498 to 598 nm bandpass filter. Fields were randomly selected, and those with similar cell densities were imaged. To observe individual mitochondria, z-stack images were taken in a series of 17 slices per cell with a thickness of 0.5 μm per slice.

Mitochondrial morphology was quantified using a computer-assisted morphometric analysis application to calculate form factor (FF) and aspect ratio (AR) values as described elsewhere [26]. Images of mitochondria were analyzed using ImageJ software. The images were first processed with a median filter to obtain isolated and equalized fluorescent pixels. Mitochondria were then subjected to particle analysis to obtain FF values (perimeter^2^/4π * area) and AR values (length of major axis/length of minor axis).

### 4.11. Measurement of Mitochondrial Bioenergetics

The real-time measurement of the oxygen consumption rate (OCR) was performed using a Seahorse XFe24 extracellular flux analyzer (Agilent Technologies, Santa Clara, CA, USA). Cells were seeded at a concentration of 20,000 cells/well in XFe24 microplates (Agilent Technologies) and treated with the corresponding medium for 48 h. Prior to OCR measurement, the medium was replaced with prewarmed Seahorse XF DMEM medium (103335-100, Agilent Technologies) containing 5.5 mM glucose (G6152, Sigma-Aldrich, Saint Louis, MO, USA), 1 mM sodium pyruvate (11360070, Gibco), and 4 mM L-glutamine (25030081, Gibco), and the cells were incubated at 37 °C in a non-CO_2_ incubator for 60 min to allow the temperature and pH to reach equilibrium. The OCR was then measured under basal conditions, and after serial injection of 10 µM oligomycin (11341, Cayman Chemical, Ann Arbor, MI, USA), 100 µM carbonylcyanide-4-(trifluoro-methoxy) phenylhydrazone (FCCP; 15218, Cayman Chemical), and a mixture of 5 µM rotenone (13995, Cayman Chemical) and 5 µM antimycin A (A8674, Sigma-Aldrich) to determine the values for the basal respiration, adenosine triphosphate (ATP) production, proton leak, maximal respiration, spare respiratory capacity, and non-mitochondrial oxygen consumption. After the assays, the cells in each well were lysed with RIPA lysis buffer on ice, and the protein concentrations were determined using a Pierce^TM^ BCA gold protein assay (Thermo Scientific) according to the manufacturer’s guidelines for normalization of the OCR values.

### 4.12. Statistical Analysis

The data represent results from at least three biological replicates and are presented as means ± SEM. An independent sample *t*-test or Mann–Whitney U test was used for comparisons between the two groups using JASP (v0.13, Amsterdam, The Netherlands). The significance level was set at α < 0.05.

## 5. Conclusions

Our study revealed that increased ROS levels and mitochondrial dysfunction occurred in 661w cells after incubation in a high-glucose concentration. The observed changes may underly the causes of neural cell death during the development of DR. Diabetes-induced ROS production has been suggested to compromise the function of the electron transport chain and damage mitochondrial DNA [75]. In addition, mitochondrial dysfunction has been associated with insulin-secreting pancreatic β-cell loss in triggering diabetes [76] and vascular endothelial damage due to the accumulation of redox-sensitive gelatin matrix metalloproteinases in DR [77]. Therefore, a pharmaceutical agent with mitochondrial stabilizing and antioxidative properties might be a suitable agent for treatment and prevention of further neurodegeneration in DR.

## Figures and Tables

**Figure 1 ijms-23-13366-f001:**
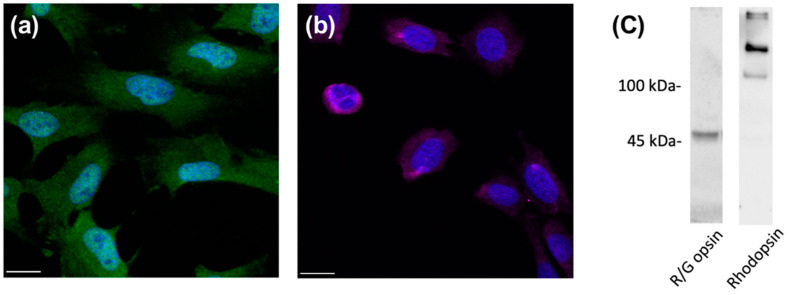
Expression of photoreceptor-specific markers in 661w cells. Immunocytochemical staining of 661w cells for (**a**) red/green opsin (green) and (**b**) rhodopsin (violet). DAPI (blue) was used to counterstain the nuclei. Scale bar: 20 µm. Immunoblot analysis (**c**) was used to confirm the expression of red/green opsin (~48 kDa) and rhodopsin (oligomeric form, >100 kDa) in 661W cells.

**Figure 2 ijms-23-13366-f002:**
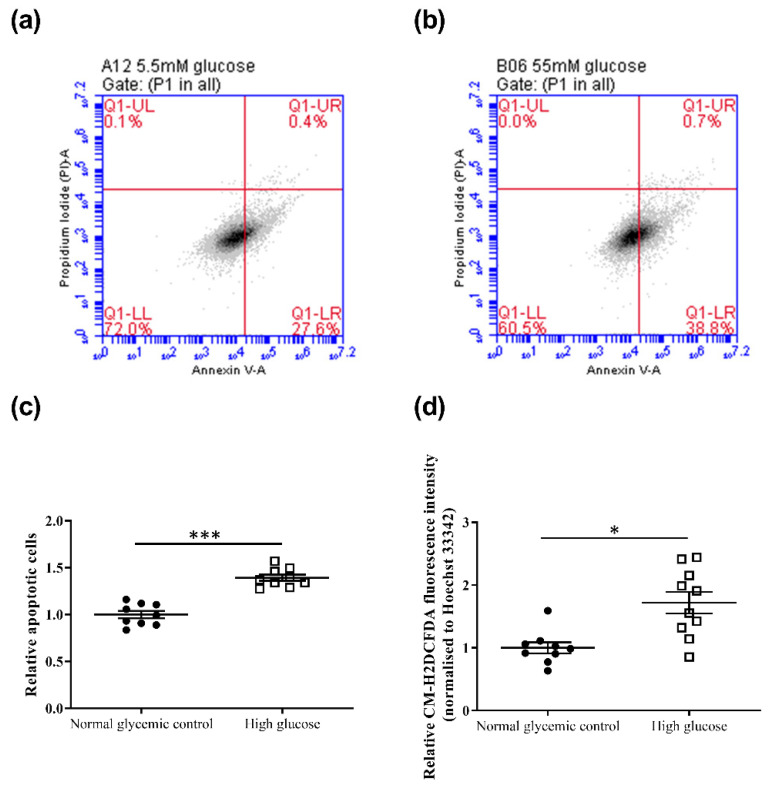
Effect of high-glucose apoptosis and ROS levels on the 661w cell line. Representative flow cytometry dot plots of cells incubated in (**a**) normal glycemic control and (**b**) high glucose for annexin V-PI counterstain. (**c**) Dot plots showing the relative change in cell apoptosis of 661w cells under the two conditions. (**d**) Relative change in ROS levels (CM-H2DCFDA fluorescence intensity normalized to Hoechst 33342 fluorescence intensity) of 661w cells in normal glycemic control and high glucose. (* *p* < 0.05, *** *p* < 0.001; data presented as mean ± SEM. n ≥ 9 (biological replicates)).

**Figure 3 ijms-23-13366-f003:**
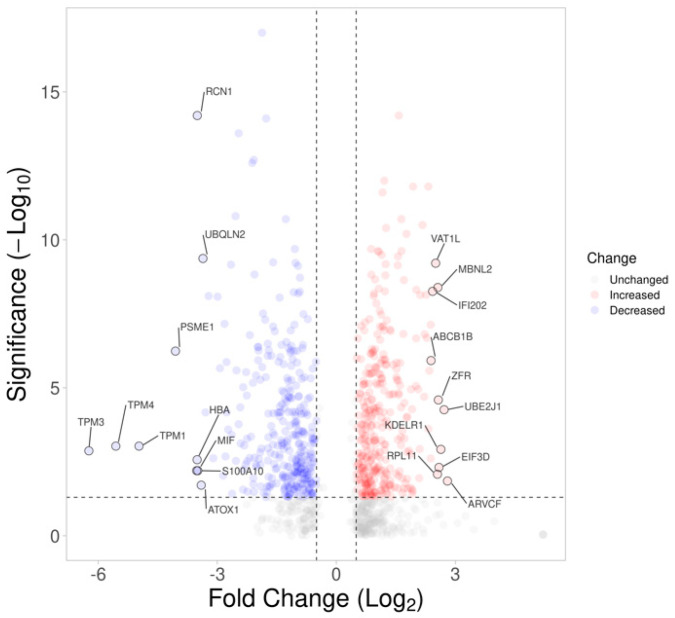
Volcano plot displaying the statistical *p*-value with the magnitude of fold change between proteins of 661w cells after 48 h incubation under high glucose vs. a normal glycemic control. The non-axial vertical dashed lines denote a fold change, of ±0.58 Log2 unit (i.e., ±1.5−fold change), whereas the non-axial horizontal dashed line denotes 1.30 −Log_10_
*p*-value (i.e., *p* = 0.05), which is the significance threshold prior to logarithmic transformation. A total of 992 of 2802 proteins were found to be differentially expressed (blue: downregulated; red: upregulated). The top 10 up- and downregulated proteins are labelled.

**Figure 4 ijms-23-13366-f004:**
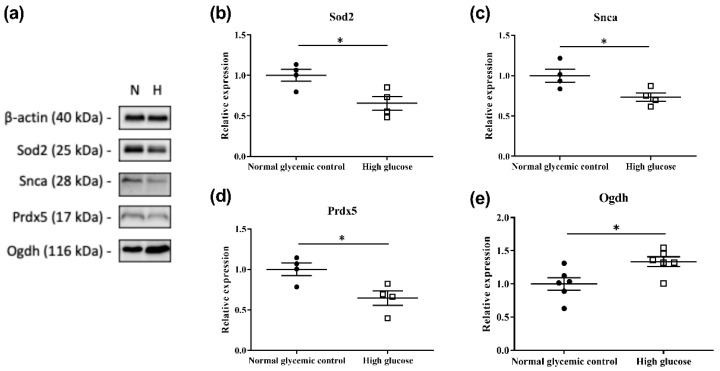
Validation of four differentially expressed proteins involved in mitochondrial dysfunction by Western blot analysis. (**a**) Representative Western blot (N: normal glycemic control and H: high glucose); (**b**–**e**) plots showing the densitometry analysis for protein levels relative to the normal glycemic control. β-actin was used as a loading control. Results illustrate the downregulation of Sod2 (~25 kDa), Snca (dimer form, ~28 kDa), and Prdx5 (~17 kDa) and the upregulation of Ogdh (~116 kDa) in 661w cells incubated in high glucose, which are consistent with our MS results. (* *p* < 0.05; data presented as mean ± SEM. n ≥ 4 (biological replicates) for each plot).

**Figure 5 ijms-23-13366-f005:**
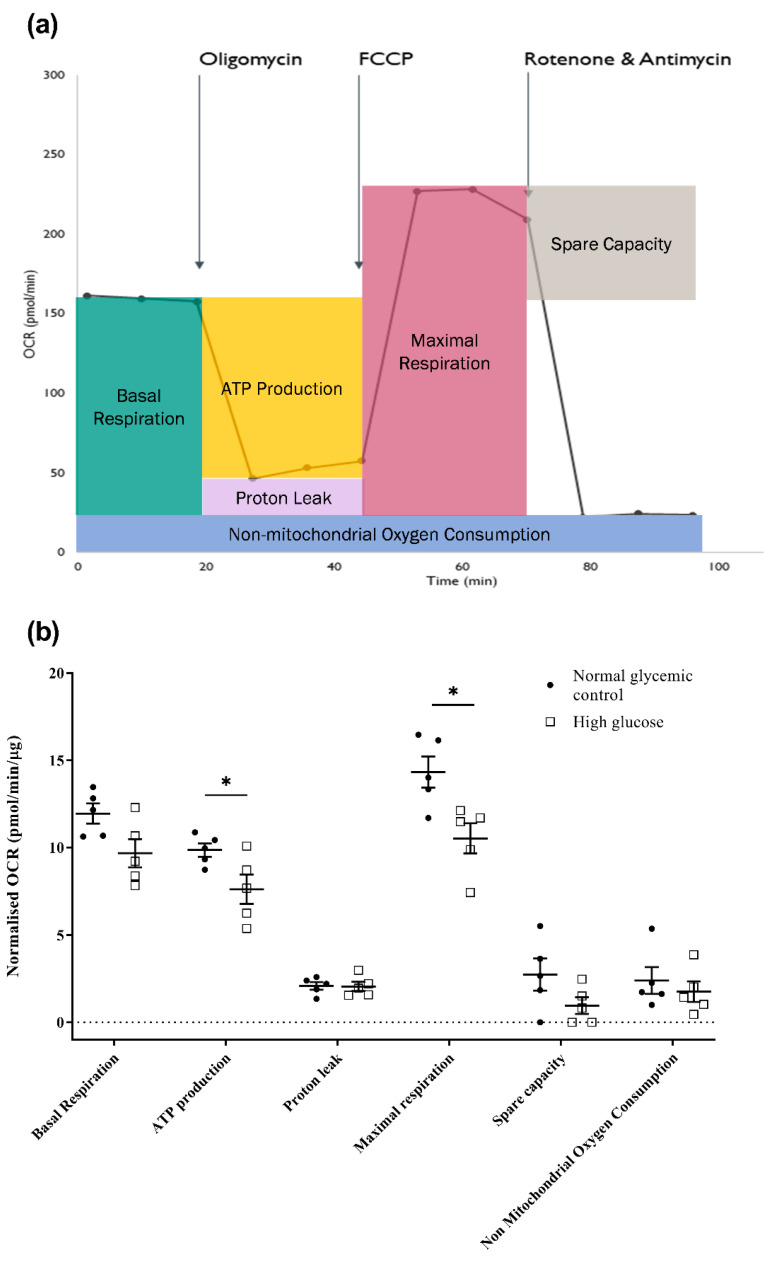
Effect of high glucose on mitochondrial bioenergetics of 661w cells. (**a**) Illustrative diagram of how key mitochondrial function parameters are divided from the Seahorse Mito Stress assay. The Seahorse XF analyzer first measures the oxygen consumption rate (OCR) as the basal respiration of the cells, and the rate of mitochondrial ATP synthesis can be estimated from the decline in OCR after adding oligomycin, an ATP synthase (complex V) inhibitor, with the remaining OCR representing the proton leak across the mitochondrial membrane in situ. The subsequent addition of an uncoupling agent, FCCP, collapses the proton gradient and disrupts the mitochondrial membrane potential, allowing the oxygen consumption of complex IV, the rate-limiting step of oxidative phosphorylation, to reach the maximum levels. This increase in OCR reflects the maximal respiration rate of the mitochondria, and the difference between the maximal respiration and basal respiration indicates the spare capacity of the mitochondria. Moreover, the injection of rotenone and antimycin A mixture ceases mitochondrial respiration, and the remaining OCR reveals the non-mitochondria-related usage of oxygen. (**b**) Dot plots showing the effect of high glucose on the key mitochondrial function parameters of 661w cells. (* *p* < 0.05. Data presented as means ± SEM. n = 5 (biological replicates)).

**Figure 6 ijms-23-13366-f006:**
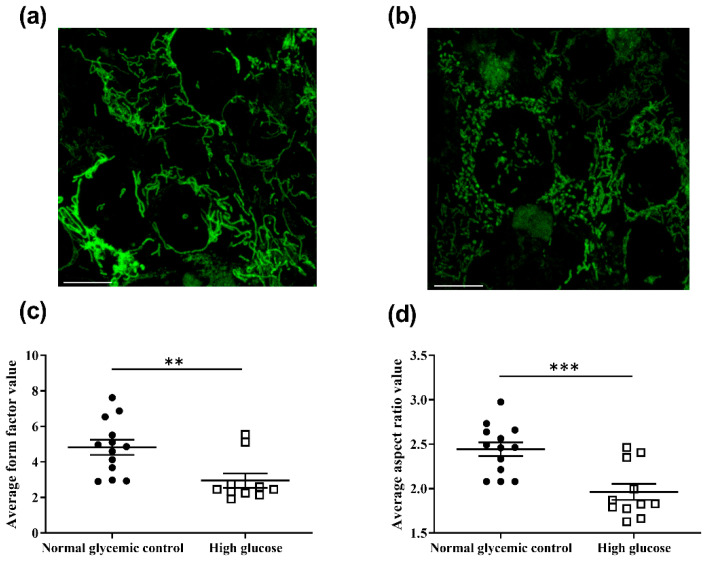
High-glucose-induced mitochondrial morphology changes and increased Fis1 levels and cytochrome c release in 661w cells. Representative confocal images showing mitochondrial morphology of 661w cells incubated in (**a**) the normal glycemic control and (**b**) high glucose. The scale bar represents 10 μm. Mitochondrial fragmentation was observed in 661w cells incubated with high glucose concentration. Graph showing the mean (**c**) form factor (FF) and (**d**) aspect ratio (AR) values for mitochondria of 661w cells after 48 h incubation in the two glucose concentrations. FF indicates the mitochondrial length and degree of mitochondrial branching. An FF value of one reflects a circular, unbranched mitochondrion, whereas a higher FF value implies a longer, more branched mitochondrion. AR represents the circularity of a mitochondrion, with a value of one indicating a perfect circle; the AR value increases with elongated and elliptical mitochondria. (** *p* < 0.01 *** *p* < 0.001; data presented as mean ± SEM. n ≥ 9 (biological replicates)). (**e**) Representative Western blot and (**f**) graphic illustration showing the Fis1 expression in 661w cells after normalization to the actin signal relative to the normal glycemic control. (** *p* < 0.01; data presented as mean ± SEM. n = 3 (biological replicates)). Cytochrome *c* translocation to the cytosol was assessed by Western blot analysis after subcellular fractionation. Cytosolic protein fractions were isolated from 661w cells after 48 h incubation under normal glycemic control and high-glucose conditions. (**g**) Representative image of Western blot analysis. (**h**) Graphic illustration showing the relative change in cytochrome c levels in the cytosol of 661w cells under high glucose after normalization to the actin signal (* *p* < 0.05; data presented as mean ± SEM. n = 3 (biological replicates) for each condition).

**Figure 7 ijms-23-13366-f007:**
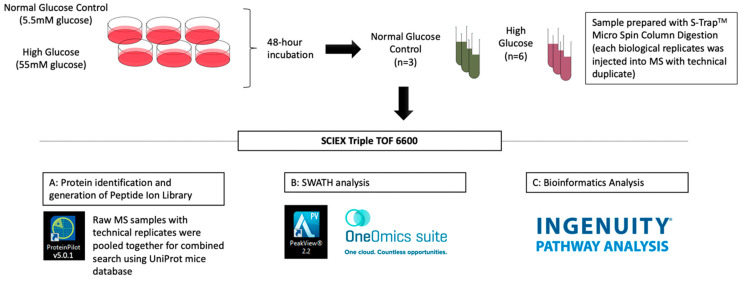
Schematic flowchart of the liquid chromatography-tandem mass spectrometry (LC-MS/MS) experiment of 661w cells incubated in high glucose vs. the normal glycemic control.

**Table 1 ijms-23-13366-t001:** The top five significant pathways revealed by pathway analysis using the IPA and their associated *p*-values (Fischer’s exact test) and z-scores. The z-score represents activation or suppression of the corresponding pathway, with “NaN” denoting unpredicted.

		*p*-Value	z-Core
1	Protein Ubiquitination Pathway	1.08 × 10^−14^	NaN
2	Mitochondrial Dysfunction	4.68 × 10^−13^	NaN
3	Oxidative Phosphorylation	6.84 × 10^−11^	−2.828
4	Epithelial Adherens Junction Signaling	3.15 × 10^−8^	−1.789
5	Actin Cytoskeleton Signaling	3.34 × 10^−8^	−0.229

**Table 2 ijms-23-13366-t002:** Differentially expressed proteins related to mitochondrial dysfunction from the SWATH experiment.

Protein Name	UniProt Accession	Gene Name	Fold-Change	*p*-Value
ATP synthase F1 subunit delta	Q9D3D9	*atp5f1d*	0.26	5.49 × 10^−7^
ATP synthase membrane subunit f	P56135	*atp5mf*	3.09	4.69 × 10^−4^
ATP synthase membrane subunit g	Q9CPQ8	*atp5mg*	3.76	3.89 × 10^−2^
ATP synthase peripheral stalk subunit d	Q9DCX2	*atp5pd*	0.33	5.70 × 10^−10^
Catalase	P24270	*cat*	0.58	1.11 × 10^−2^
Cytochrome c oxidase subunit 4I1	P19783	*cox4i1*	0.34	3.98 × 10^−4^
Cytochrome c oxidase subunit 6C	Q9CPQ1	*cox6c*	2.32	1.11 × 10^−6^
Cytochrome b5 type B	Q9CQX2	*cyb5b*	0.11	8.02 × 10^−9^
Cytochrome c1	Q9D0M3	*cyc1*	0.33	3.32 × 10^−4^
Cytochrome c, somatic	P62897	*cycs*	0.51	7.11 × 10^−3^
NADH:ubiquinone oxidoreductase subunit A8	Q9DCJ5	*ndufa8*	0.45	7.92 × 10^−3^
NADH:ubiquinone oxidoreductase core subunit S1	Q91VD9	*ndufs1*	0.52	8.00 × 10^−4^
NADH:ubiquinone oxidoreductase core subunit S2	Q91WD5	*ndufs2*	0.51	6.42 × 10^−9^
NADH:ubiquinone oxidoreductase core subunit S3	Q9DCT2	*ndufs3*	0.55	4.61 × 10^−7^
NADH:ubiquinone oxidoreductase subunit S4	Q9CXZ1	*ndufs4*	0.51	7.52 × 10^−10^
Oxoglutarate dehydrogenase	Q60597	*ogdh*	1.60	7.09 × 10^−3^
Parkinsonism associated deglycase	Q99LX0	*park7*	0.36	9.07 × 10^−4^
Pyruvate dehydrogenase E1 subunit alpha 1	P35486	*pdha1*	2.35	1.67 × 10^−6^
Peroxiredoxin 5	P99029	*prdx5*	0.39	9.49 × 10^−3^
Succinate dehydrogenase complex iron sulfur subunit B	Q9CQA3	*sdhb*	0.39	3.52 × 10^−5^
Synuclein alpha	O55042	*snca*	0.57	4.36 × 10^−6^
Superoxide dismutase 2	P09671	*sod2*	0.34	1.83 × 10^−3^
Ubiquinol-cytochrome c reductase binding protein	Q9D855	*uqcrb*	0.21	5.22 × 10^−4^
Ubiquinol-cytochrome c reductase core protein 1	Q9CZ13	*uqcrc1*	0.56	2.91 × 10^−2^
Ubiquinol-cytochrome c reductase core protein 2	Q9DB77	*uqcrc2*	0.54	5.35 × 10^−3^
Ubiquinol-cytochrome c reductase, Rieske iron-sulfur polypeptide 1	Q9CR68	*uqcrfs1*	0.54	1.05 × 10^−3^

## Data Availability

The data presented in this study are available upon request from the corresponding author.

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
