# Peer review of "Proteomic Profiling Revealed Mitochondrial Dysfunction in Photoreceptor Cells under Hyperglycemia"

_ijms, 2022, doi:10.3390/ijms232113366_

Round 1

Reviewer 1 Report

This is an interesting paper where the authors us state of the art techniques to determine the effect of high glucose conditions on photoreceptor cells. However, there are some concerns that the authors have to address before publishing this

Results

2.2: On line 92, you have 83.1 4.2% and 74.3 9.3%. I believe this is a SE, and missing the +/- sign

In the figure 2, you have FITC-A and PE-A labels for the flow cytometry plots. This is not the right labels for a manuscript. You have to list annexin V and PI as the labels.  

Please use dot plots instead of bar graphs for panels c and d.

2.3: The main takeaway message from a volcano plot, is to highlight the data points that are not only significantly different in p value, but also show a drastic change in fold change. While I see some very interesting points that are altered in both these aspects in figure 3 (for example, 3 points are showing a Log 2 fold change of -6), you do not mention what any of these proteins are.  You just highlight all the proteins that are significantly different, and hence the readers do not get to see the proteins that are of utmost importance.

2.4: I do not see any difference in Snca expression, yet you see a significant reduction in the bar graph (30-40%). Also Ogdh shows only a small difference. Please use representative blots. Also, again use only dot plots. Please change all the bar graphs to dot plots from here on in. 

2.5: Please explain what the mitochondrial poisons do and what each one means in reference to the inhibition of specific complexes. Also does the result from the seahorse tell you which complexes were specifically affected by high glucose? Did you use substrates for specific complexes?

2.6: Why are figures here divided up into figure 6, 7 and 8. All of them are giving evidence of mitochondrial damage, so please include them in different panels of figure 6. To have one blot of 2 lanes for a complete figure, seems meaningless by itself.

Importantly, why do you see an increase in ROS when your OXPHOS proteins are downregulated. Previously, an increase in OXPHOS proteins were observed with high glucose treatment for endothelial cells, and also increase in ROS production with increased cell apoptosis (Haspula et al., 2019; Front Phy). This is a known mechanism. Also observed was a shift from glycolysis to OXPHOS. So are you implying that since OXPHOS is low, there is a possible over-reliance on glycolysis for the cell's ATP needs? Please refer to the aforementioned paper for more information. 

Reviewer 2 Report

This manuscript describes a proteomic study aiming to reveal how high glucose treatment could affect global protein levels of photoreceptor cells. First, the 661w cell line was selected and validated as a model to represent the primary photoreceptor cells. Using the 661w cell line, the authors found that high glucose treatment induced apoptosis without significantly decreasing cell viability. Then, the authors performed SWATH-based quantitative proteomics and identified 625 differentially expressed proteins. Among them, the authors focused on proteins involved in mitochondrial dysfunction and demonstrated that high glucose treatment resulted in a significant reduction in ATP production and maximal respiration in 661w cells, which was accompanied by mitochondrial morphology aberrations. Finally, the authors found an increase in cytochrome c translocation from mitochondrial to the cytosol, suggesting that the high glucose-induced increase in apoptosis may result from mitochondrial-mediated pathways. The findings presented in the manuscript could contribute to a better understanding of the mechanisms of diabetic retinopathy.

Overall, the experiments in this study are well-designed and well-documented in the methods sections. I recommend publication upon addressing the concerns detailed below in a revised manuscript.

Minor points:

1. The panels (a) and (b) in Figure 2 should be better annotated to reflect the identities of the cell population in each region. In addition, the axis labels should include the corresponding antibodies (e.g., Annexin V – FITC for the x-axis). The current figures make little sense for readers not familiar with Annexin V/Dead cell apoptosis assay.

2. The protein level differences between normal and high glucose groups in Figure 4 (a) are rather subtle. The densitometry analysis results will be sensitive to the exact data processing methods, which should be documented in the methodology section in more detail. For instance, were the western plot protein signal normalized to the corresponding beta-actin band in each group?

3. In Figure 7, the Fis1 appeared as a 68 kDa band instead of 17 kDa. The authors suggested this may result from Fis1-Mief1 complex formation, which is unusual under denaturing gel electrophoresis. This result should be cross-validated with Fis1 antibodies from different vendors and/or anti-Mief1 antibodies.

Round 2

Reviewer 1 Report

Thank you for making the changes I asked for. The paper reads better and figures look much improved.

Just one minor correction, You still have bar graphs for Figure 5b. Please change this to dot plots
